

# Evaluating the atmospheric drivers leading to the December Flood 2014 in Schleswig–Holstein, Germany

Nils H. Schade[1]

[1]Federal Maritime and Hydrographic Agency (BSH), Hamburg, D–20359, Germany

*Correspondence to*: Dr. Nils H. Schade (nils.schade@bsh.de)

**Abstract.** Regional analyses of atmospheric conditions that may cause flooding of important transport infrastructure (railway tracks, highways/roads, rivers/channels) and subsequent adaptation measures are part of the Expertennetzwerk initiated by the German Federal Ministry of Transport and Digital Infrastructure (BMVI). As an exemplary case study, the December flood 2014 in Schleswig–Holstein, Germany, was investigated. Atmospheric conditions at the onset of the flood event are described and evaluated with respect to the general weather situation, initial wetness, and event precipitation. Predominantly persistent westerly situations directed several low pressure systems over the North Sea to Schleswig–Holstein during December 2014, accompanied by prolonged rainfall and finally a strong event precipitation in southern Schleswig–Holstein causing several inland gauges to exceed their by then maximum water levels. An additional storm surge hindering drainage of the catchments into the North and Baltic Sea could have been fatal. Results show that the antecedent precipitation index (API) is able to reflect the soil moisture conditions and, in combination with the maximum 3–day precipitation sum (R3d), to capture the two main drivers finally leading to the flood: (1) Initial wetness of north western Schleswig–Holstein, and (2) strong event precipitation in southern and eastern Schleswig–Holstein from 21–23 December while both indices exceeded their respective 5–year return periods. Further, trend analyses show that both API and R3d are increasing while regional patterns match the north eastward shift of cyclone pathways during recent years, leading to higher risk of flooding in Schleswig–Holstein. Within the Expertennetzwerk, investigations of these and further indices/drivers for earth system changes (e.g. wind surge, sea level rise, land cover changes, and others) derived from observations, reanalyses, and regional climate model data are planned for all German coastal areas: Results can be expected to lead to improved adaptation measures to floods under climate change conditions wherever catchments have to be drained and infrastructures and ecosystems may be harmed, e.g. in other Baltic Sea regions.

## 1 Introduction

In Dezember 2014, predominant westerly general weather situations (GWSs) caused a Major Baltic Inflow (MIB) event (see e.g. Lehmann et al., 2016; Post and Lehmann, 2016). At the same time, persistent rainfall in combination with an extreme precipitation event during the Christmas Holidays led to the flooding of several catchment areas in Schleswig–Holstein,



Germany, located between the North and Baltic Sea. Both events mark exemplary atmospheric and hydrologic responses within the causality chain illustrating the importance of interdisciplinary research in this area. In this regard, the region Schleswig–Holstein is a potent "blue spot" dealing with multiple drivers for earth system changes in the North and Baltic Sea region. It is affected in many ways by extremes, especially under climate change conditions: (1) Considerable areas in

the southern parts lie beneath sea level and have to be drained artificially, (2) long lasting and heavy rainfall events lead to increased flooding possibility of economically relevant parts of the country. The North and Baltic Sea Channel (NOK, http://www.wsa-kiel.wsv.de/Nord-Ostsee-Kanal) for example, also known as "Kiel Canal", is the most important waterway in this region. In fact, with over 30,000 passages per year, it is the busiest artificial waterway worldwide (e.g. Lübbecke et al., 2014). But the NOK is not only important for transportation; it also serves as drainage of several catchments, e.g. the

upper Eider basin, while the water level has to be regulated within a few decimetres to keep shipping traffic possible. Therefore, the atmospherical and hydrological conditions have to be monitored carefully concerning extremes and changes thereof.

The physical conditions of the North Sea are the dominant factor controlling both meteorology and hydrology in European coastal regions (see e.g. Attema and Lenderink, 2014). Of particular importance are the actual wind and water levels –

15 including future sea level rise – and the predominant GWS. According to Randall et al. (2007), large scale and prolonged extreme events result from a persistent GWS in conjunction with the interaction between air and sea (air and soil, respectively). These interactions are of particular importance for coastal areas. Hydrological extremes, like flooding events, thereby are rather caused by unusual and unfavourable combinations of different influencing factors than by extremes of these factors themselves (Klemes, 1993). For instance, storm surges in combination with heavy but not extreme rain falls

may lead to problematic drainage situations due to high seaward water levels (see e.g. Wahl et al., 2015). According to investigations by Kew et al. (2013) for the Rhine delta, the probability of extreme surge conditions following extreme 20–day precipitation sums is even 3 times higher than estimated from treating extreme surge and discharge probabilities as independent. Also, a combination of initial catchment wetness and a single heavy, yet not extreme, precipitation event alone may lead to flooding. Berthet et al. (2009) and Pathiraja et al. (2012) show that catchment wetness actually is a crucial

parameter in flood forecasting. Further, given the difficulties in estimating the catchment wetness arising from inadequate records of soil moisture conditions (e.g. Albergel et al, 2013), Woldemeskel and Sharma (2016) point out the role of antecedent precipitation as a surrogate variable for any flood assessment under global warming conditions.

In the following, the observed situation, predominant GWS, and precipitation indices describing soil moisture condition and event precipitation are investigated for the December Flood 2014 in Schleswig–Holstein, Germany. An extensive evaluation

concerning the hydrology based on catchment gauge data has already been undertaken by the Landesbetrieb für Küstenschutz, Nationalpark und Meeresschutz Schleswig–Holstein (LKN–SH) and the Landesamt für Landwirtschaft, Umwelt und ländliche Räume Schleswig–Holstein (LLUR–SH) in a separate report (LKN–SH and LLUR–SH, 2015). Therefore, the focus of this paper lies on the atmospheric conditions. The aim is to show that the method of Schröter et al. (2015) to classify nationwide flood events can be applied on a regional scale and that the indices used, namely the antecedent



precipitation index (API) and 3–day event precipitation sum (R3d), provide useful information about changing local flood regimes in a warming climate. Within the Expertennetzwerk of the German Federal Ministry of Transport and Digital Infrastructure (BMVI, http://www.bmvi-expertennetzwerk.de), all methods used in this paper will be applied to reanalyses and (regional) climate model data as well. The results can expected to be of great value for the work in national and
international projects dealing with adaptation of transport and infrastructure under future climate change.

The remainder of this paper is structured as follows: At first, data and methods (chapter 2) are described. An evaluation of the atmospheric drivers leading to the December flood 2014 and a discussion of the findings are offered in chapter 3: Therein, the observational basis is presented in chapter 3.1, the statistical analysis of the precipitation indices following the method of Schröter et al. (2015) in chapter 3.2, as well as a trend analysis of the indices investigated. Finally, concluding
remarks are given in chapter 4.

## 2 Data and Methodology

### 2.1 General Weather Situation

Two different General Weather Situation (GWS) classification methods were compared to describe the situation in SH during December 2014: (1) The modified Lamb Weather Types (LWT, Jenkinson and Collison, 1977) used at BSH (Löwe et
al., 2005) with a model centre over the central North Sea, and (2) the objective classification (OWTC; Dittmann et al., 1995; Bissolli and Dittmann, 2001) of the German Meteorological Service (DWD) with a model centre over central Germany. Further differences are due to the input parameters: While LWT is based solely on sea level pressure data, here NCEP/NCAR Reanalysis 1 (Kalnay et al., 1996), at 16 grid points over northern Europe, OWTC input data include air pressure, temperature, wind, and water vapour content on different height levels derived from the current operational GME
(Global Model Extended) of DWD (http://www.dwd.de/EN/ourservices/wetterlagenklassifikation/wetterlagenklassifikation). Further, OWTC output parameters include cyclonality on two height levels (950 and 500 hPa) and a humidity index ("wet" and "dry") that describes the precipitable water content of the atmosphere compared to the long term daily mean. LWT output, however, includes a gale index in four categories (from "no gale" to "very severe gale") derived from the strength of the geostrophic flow and the vorticity.

### 2.2 Precipitation and Moisture Indices

Schröter et al. (2015) have investigated and ranked 76 nationwide flood events concerning their severity and affecting at least 10 % of the German river catchments over a period from 1960 to 2009. Further included were the floods from 1954 and 2013 (Blöschl et al., 2013). The investigations based on the dataset from Uhlemann et al. (2010) using time series of daily mean discharge records at 162 gauge stations of the German Water and Shipment Administration (WSV) and the German
Federal Institute of Hydrology (BfG). Additionally, Schröter et al. (2015) used the REGNIE data set (see e.g. Rauthe et al., 2013) with a spatial resolution of 1 x 1 km provided by DWD to describe the meteorological situation of these events. Basic





idea using this approach is the assumption that a combination of extreme initial wetness (i.e. oversaturation of the soil) and a strong but not extraordinary event precipitation leads to flooding. These factors were evaluated by means of two indices: (1) The maximum 3–day precipitation sum (R3d) as trigger of the flood, calculated at each grid point separately within a window of ± 10 days around the onset of the flood event, and (2) the initial antecedent precipitation index (API), calculated

from the sum of daily precipitation at each grid point $R_i(x, y)$ and weighted with respect to the time span ($m = 30$ days) of rainfall occurrence prior to the R3d to assure a clear separation of both indices, see Eq. (1):

$$API(x,y) = \sum_{i=1}^{30} k^i \, R_i(x, y, (m - i)). \qquad (1)$$

Here, $i$ marks the day prior the R3d and $k = 0.9$ a depletion constant that approximates the decrease of soil moisture due to evapotranspiration and percolation to deeper soil layers. Using this approach, the rainfall at day one prior the R3d is

weighted highest.

Both indices were calculated for the December Flood 2014. The constant $k$ was not changed; however, future investigations could include regional soil types at a high resolution (if accessible). Further, it should be noted that the coastal regions were excluded by Schröter et al. (2015) since floods might be affected by the water level conditions in the North and Baltic Sea, i.e. the possibilities for drainage (personal communication K. Schröter). Since sea gauge data did not show any extremes and

drainage was possible at all times during the December Flood 2014, a comparative analysis is justified and might help to point at regional risk potentials, even for spatially limited flood events.

Schröter et al. (2015) further defined indices to classify the severity of each event $k$ investigated as aggregated measure $S_X^k$ for each parameter $X$, in this case, the event precipitation (P) and the initial wetness (W): All values $X_{x,y}^k$ ($X$ here stands for either R3d or API) at each grid point $(x, y)$ that exceeded the values of the respective 5y return period were divided by the

latter, their ratios summed up, and finally normalized by the number of REGNIE grid points $\Gamma$ in Germany, following Eq. (2):

$$S_X^k = \frac{100}{\Gamma} \sum_{x,y} \left\{ \frac{X_{x,y}^k}{X_{x,y}^{5 \, y \, RP}} \right\} \; \Big| \; X_{x,y}^k \geq X_{x,y}^{5 \, yr \, RP} \qquad (2)$$

Now, all events can be ranked and compared. The floods in June 2013 (see Belz et al., 2013; Stein and Malitz, 2013, Belz et al., 2014), holding as the heaviest and severest event in the last 60 years, and July 1954 were special in the sense that both

represent the extremes in case of the severity of event precipitation P (July 1954, P = 55.2) and initial wetness W (June 2013, W = 114.1).

For the following investigations REGNIE was used as well and, further, the Matlab toolbox WAFO (WAFO–group, 2000) for the statistical evaluation of the extreme precipitation indices. According to Schröter et al. (2015), the yearly maximum 3– day precipitation sums and the respective 30–day antecedent precipitation were calculated at each REGNIE grid point. Then,

5–year return periods (5yrRPs) were derived at each grid point using the Gumbel distribution over the base period 1960– 2009 to compute the severity indices.



## 2.3 Trend Analyses

Mean trends at the 95 % significance level for the five highest R3d (R3dfivemax) and API (APIfivemax) values per year were calculated over 30–year running intervals from 1960–89 to 1985–2014 for the Kiel Canal catchment (EZG NOK), Schleswig–Holstein (SH), and all of Germany (D). Instead of the yearly maximum alone, the five highest events per year

were chosen as to obtain more reliable and robust statistics. A modified version of the Mann–Kendall test (see Hamed and Rao, 1998) was used to determine significant trends avoiding misleading results due to autocorrelation (in case autocorrelation is greater than zero). All trends were calculated at each grid point separately. Then, area means were derived.

## 3 Results and Discussion

### 3.1 Observed Situation

The December 2014 was predominated by westerly GWSs lasting for several weeks. Therefore, a number of low pressure systems were led from the North Atlantic over northern Europe in short progression. Exemplarily, the systems ALEXANDRA and BILLIE (11/12/2014) both characterized by wet maritime air and stormy conditions with gusts from 8 to 10 Bft observed all over Schleswig–Holstein (SH) and Hamburg (HH) are shown in Figure 1.

#### 3.1.1 General Weather Situation (GWS)

In general, both classification methods show predominant westerly GWSs from 5 December onwards with north westerly (NW) situations during the heavy precipitation event from 21–23 December (Table 1): OWTC shows humid conditions, LWT "gale". However, differences are apparent during the first precipitation event: The cores of the low pressure systems are centred far north, categorized by LWT as "severe gale" (ALEXANDRA) and "gale" (BILLIE) with south westerly (SW) cyclonic flow (Fig. 2a/b). OWTC on the other hand categorized a NW anticyclonic flow and dry conditions. An explanation

provides the respective model centre; OWTC is focused over central Germany while LWT is ideally centred in the North Sea. Since most of southern and central Germany was unaffected by this precipitation event, most of the model domain was indeed "dry".

It becomes obvious that the use of the classification method is subjected to several factors. Amongst them, the method should be suitable to the region of interest and capture its unique features. Here, the LWT seem more appropriate.

Nevertheless, both GWS clearly show that not only one weather type but the succession of similar (westerly) types was important to the overall high soil moisture conditions, i.e. in generating prolonged rainfall, especially in northern Schleswig–Holstein. Additionally, the extreme precipitation event in southern Schleswig–Holstein was caused by a succession of NW types from 19–23 December (5 days, LWT) and 17–23 December (7 days, OWTC), respectively. Considering the mean life time of the NW type of 1.82 days (base period 1971–2000, Löwe et al., 2013, Table 2–10), the event was extraordinary for

this region.



### 3.1.2 Precipitation

Above average monthly precipitation amounts between 80 and 160 mm were recorded at the German coasts during December 2014; local monthly means were exceeded by more than double that values and old records were broken. In SH, values of 175 mm up to 225 mm were reached (Fig. 3a) which corresponds to about 225–300 % of the long term means (Fig. 3b). All over the rest of Germany, the December 2014 was unremarkable with maximum mean values around or clearly below those of the reference period (1961–1990).

Looking at daily precipitation sums from the REGNIE data set, two main rainfall periods can be distinguished: One from 10–12 December, more pronounced in northern SH, and one from 18–24 December (Fig. 4). Maximum daily precipitation was detected from 22–23 December in southern SH and HH with local values exceeding 50 mm corresponding to the standard monthly mean values.

As seen in Figure 4(a–c), the first rainfall period begins in the far north eastern (NE) SH on 10 December slowly progressing to the south. It further shows that not only SH was affected during this event: Pronounced rainfall was detected north of the Eifel region on 12 December. Figure 4(d–f) displays the main precipitation event from 21 to 23 December. Now, mainly northern Germany is affected, especially southern SH on 22 December. Values are comparable to those from selected DWD stations (LKN–SH and LLUR–SH, 2015, their Fig. 1) showing that REGNIE performs well.

### 3.1.3 Soil Moisture

Additional investigations using modelled soil moisture data from DWD's Agrometeorological Research Centre (ZAMF) for sandy loam soil and cultivation with sugar beets show highest values in the SH region with up to 139 % nFK in the north for 21 December 2014 (start date of the corresponding event precipitation). Values are decreasing southward, but never below 100 % nFK except for the south of SH (Fig. 5a). The unit [% nFK] describes the saturation in percent effective field moisture capacity of the upper 60cm of soil. If soil moisture exceeds 100 % nFK, the actual water content is higher than usable for plants (DWD, 2016), i.e. most of northern and central SH at the onset of the main precipitation event. The south to north gradient is in accordance with precipitation data showing a slow progression of rainfall events from north to south (see chapter 3.1.2).

ZAMF also provides soil moisture data for loamy sand soil and cultivation with winter grain. Using this data, values in northern and central SH are between 105 and 110 % nFK for the same date (Fig. 5b), only not that differentiated. It should be noted that neither the actual soil differentiation nor the degree of sealing is part of the model chain, and locally, this might be of importance (see e.g. Apel et al., 2016). Nevertheless, both soil types show the same oversaturated regions in SH with some minor differences in the Fehmarn area (eastern SH).





### 3.1.4 Gauge Data

The LKN–SH and LLUR–SH (2015) report points out that more than a third (66 out of 184) inland gauges in SH exceeded the up to date Highest High Water level (HHW) during the December Flood 2014 (see their Fig. 27). All of these gauges are located in areas affected by high event precipitation and/or high antecedent precipitation (see following chapter 3.2). Further, more than 80 % exceeded the Mean High Water level (MHW), while gauges not reaching the MHW were mainly sea gauges located in the North Sea. Return periods of half a year were hardly exceeded here (personal communication). Therefore, the December Flood 2014 could have been much worse if an additional storm surge would have hindered the drainage of the SH catchments into the North Sea.

### 3.2 Precipitation Indices

### 3.2.1 Event Precipitation – R3d

Figure 6 shows the 3–day precipitation sum R3d for the December Flood 2014 in Schleswig–Holstein (Fig. 6a) and its corresponding ratio to the 5–year return period (5yrRP, Fig. 6b). The scale for R3d is set according to Schröter et al. (2015). Clearly, the main contiguous part of the event precipitation is restricted to northern Germany with some spots in central and southern Germany. It shows rather moderate maximum values of 109 mm north of Hamburg compared to the flood in 2013 with maximum values up to 300 mm (see Schröter et al., 2015, their Fig. 5, left). These differences can be explained mainly with the origin of both events: The flood 2013 was triggered by a quasi–stationary trough over central Europe in May/June leading low pressure systems with hot and humid air masses at its flanks from SE Europe up north. Additional orographic effects caused by the mountain ridges in central Europe, large–scale uplifting downstream the low pressure systems, and embedded convective processes finally led to prolonged and extended rainfall (e.g. Belz et al., 2014). The December flood 2014 was triggered by low pressure systems with North Atlantic air masses exclusively and appeared in winter when relatively cold air cannot hold as much water.

Areas with R3d exceeding the 5yRP are centred north of Hamburg in the area of Wittenborn (see Fig. 4), the eastern NOK region, the catchments Stör and Krückau, and at the coasts of Mecklenburg–Vorpommern (Fig. 6b). Higher return periods were exceeded only locally, i.e. north of Hamburg (not shown).

### 3.2.2 Antecedent Precipitation Index – API

Figure 7 shows the corresponding values for the antecedent precipitation index API, again, scaled according to Schröter et al. (2015). Maximum values of 41.5 mm are well below those of the flood 2013 (see Schröter et al., 2015, their Fig. 7, left) and can be found in NW–SH which is in fair agreement with the soil moisture data (see Fig. 5a). In contrast to R3d, the 5yRPs for API are exceeded only in NW–SH (Fig. 7b). Higher return periods were not reached during this flood.

It is obvious that antecedent precipitation in combination with the maximum precipitation event led to SH–wide flooding in 2014 (Fig. 6/7): Areas that were struck with heavy rainfall did not need additional initial wetness to be flooded, areas with



high antecedent precipitation only small amounts of additional event precipitation. Further, it illustrates the importance of both indices to describe this flood accurately and points at potential risks in case an additional storm surge would be present during the course of a similar event, affecting the drainage of affected catchments. Nevertheless, areas that were affected the most experienced strong event precipitation in December 2014.

### 3.2.3 Severity Indices

Nationwide, the December Flood 2014 ranks with $P = 3.3$ for the event precipitation and $W = 0.6$ for the antecedent precipitation which is at least significant for the event precipitation index: It exceeds the wetness index by a factor of 5. Compared to the events in 1954 ($P = 55.2$) and 2013 ($W = 114.1$) the December Flood 2014 is of little relevance. On the regional scale, however, it affected SH as a whole (API in the northwest, R3d in the east and south) which makes this flood event particularly noteworthy (Fig. 6/7). Taking climate change into consideration, situations like this can be expected to increase in strength and occurrence. First hints can be seen in the following trend analyses conducted for API and R3d, respectively.

### 3.2.4 Trend Analyses

Figure 8 shows mean 30–year running trends for the North and Baltic Sea channel catchment area (EZG NOK), Schleswig–Holstein (SH), and all of Germany (D) for the five highest R3d and API events per year. Obviously, trends are not only highly dependent on the respective base period, showing considerable interannual variation, but on the area under investigation as well: While trends for R3d are positive in SH (with one exception) and the EZG NOK during the whole period, they become negative during recent years looking at Germany as a whole. Keeping in mind that cyclone pathways and connected extreme precipitation events are shifting north eastwards (e.g. Stendel et al., 2016), SH and the EZG NOK will unmistakably experience more and heavier extreme situations in the future. Furthermore, a clear separation of NE and SW Germany regarding significant R3d trends is evident in recent years, exemplary shown for the period 1983–2012 (Fig. 9a): Trends are positive in NE Germany and negative in SW Germany with only some local spots (e.g. mountainous areas) showing opposing trends. This also is in accordance with the shifting cyclone pathways.

API trends (Fig. 9b) are negative for all areas in the beginning, change to high positive values during the 1980ies, and settle on lower values since, but with trends to increase further. Again, D shows smallest values since the NE–SW separation is also evident but not as articulated as for R3d. Nevertheless, API as well can be expected to increase stronger in coastal areas under climate change conditions leading to wetter soil and increased risk of flooding. Combined with the higher probability of extreme precipitation events, especially for northern Germany (SH, EZG NOK), the risk increases even further.



## 4 Concluding Remarks

In the end, the December Flood 2014 in Schleswig–Holstein did not turn out as dramatic as it could have been: The flood management worked well, the infrastructure withstood the water masses for the most parts (only a few dyke breaks were reported), fire departments and voluntary aides reacted fast. The most significant damage to the transport ways appeared to

be a slope slide on freeway A1 on a distance of about 1500 m (personal communication). Nevertheless, the initial wetness in combination with strong event precipitation could have caused more severe damage. Undercutting of the railway tracks Hamburg – Kiel/Flensburg would have led to considerable restrictions for train services and transportation since this route is the main connection up north. Some fields close to the tracks were already flooded and under surveillance during the whole event. Bridges and Tunnels in the area are old and water levels up to 45 cm above the previous maximum put the

infrastructure under enormous pressure. Furthermore, an additional simultaneous storm surge could have caused severe problems, e.g. by cutting off the possibilities for drainage due to high low water levels. The meteorological situation was indeed existent: Persistent westerly weather situations with frequent low pressure systems partly classified as "gale" or even "severe gale".

The indices R3d and API used nationwide by Schröter et al. (2015) provide useful information on the regional scale as well

and give an accurate evaluation of the initial wetness and the heavy rainfall event that led to the flood in December 2014. API, especially, captures the highest soil moisture conditions modelled by ZAMF at the onset of the R3d event quite well. This is of particular interest for future evaluation of reanalyses and climate models because this method only needs precipitation data as input which makes it a cost effective estimation of the soil moisture without running additional soil models. Since catchment wetness prior to extreme precipitation events is of high importance for flood forecasts (see Berthet

et al., 2009; Pathiraja et al., 2012), API seems to be a promising surrogate, especially in case of poor observational soil moisture data (see Woldemeskel and Sharma, 2016). Nevertheless, additional high resolution information about the actual soil type, i.e. in calculating the respective depletion constant, could be an advantage. Other influencing factors/drivers like snowmelt, frost, droughts, etc. should be taken into consideration as well since each catchment exhibits its own system of dependencies (see e.g. Valiuškevičius et al., 2016).

Trend analyses indicate an increasing risk of flood prone situation in Schleswig–Holstein due to increasing R3d and API values separately or in combination over the last decades. Taking sea level rise into account (e.g. Quante et al., 2016; Wahl et al., 2013) leading to increased ground water levels and, therefore, higher initial soil moisture, flood protection and improved drainage of the affected catchments becomes even more relevant. Future work within the Expertennetzwerk will include evaluating long term changes at gauge stations in the North and Baltic Sea (Möller and Heinrich, 2016) and testing

the applicability of the above described severity indices in reanalyses and regional climate models (RCMs) since precipitation extremes are expected to increase in the future (e.g. Nikulin et al., 2010; Kharin et al., 2013; Scoccimarro et al., 2013) and the number of potentially harmful situations can be expected to increase accordingly. This holds especially true for other Baltic Sea regions, where river systems and catchments have to deal with an additional meltwater runoff.



Several other impact studies and pilot projects will investigate future planning and management of transportation under climate change scenarios, e.g. the NOK, Fehmarnsund, and coastal infrastructure. The latter may be harmed by increasing wind induced water levels in the North Sea as well (Gaslikova et al., 2012). New high resolution reanalyses like COSMO–REA6 (Bollmeyer et al., 2015) by the **H**ans–**Er**tel–**Z**entrum (HErZ), based on DWD's operational forecast model

**CO**nsortium for **S**mall–Scale **MO**delling limited–area model (COSMO–LAM; Schättler et al., 2011), will improve the hindcast evaluations and serve as input for RCM runs. In a first comparison, Kaiser–Weiss et al. (2015) have already shown advantages over global reanalyses for ground level wind data, especially in coastal and mountainous regions due to the improved spatial (6 x 6 km) and temporal (hourly) resolution. The same might be expected for the evaluation of (extreme) precipitation.

Further investigations could include extending the above described indices R3d and API to extreme and abnormal events (see Müller and Kašpar, 2014; Müller et al., 2015) including seasonality and a varying size of the catchment areas which is of particular interest for regional investigations. Also, the use of the extreme climate indices defined by the Expert Team on Climate Change Detection and Indices (ETCCDI, see e.g. Sillmann et al., 2013a; 2013b) might prove relevant.

*Acknowledgements.* The research leading to this manuscript was conducted as part of the Expertennetzwerk initiated by the

German Federal Ministry of Transport and Digital Infrastructure (BMVI). REGNIE and OWTC data have been provided by the German Meteorological Service (DWD), soil moisture data by DWD's Agrometeorological Research Centre (ZAMF). NCEP/NCAR R1 is provided by the NOAA/OAR/ESRL PSD, Boulder, Colorado, USA, from their Web site at http://www.esrl.noaa.gov/psd. Thanks go to the BSH/DWD teams for fruitful discussions and support. Personal thanks to Peter Löwe and Kai Schröter.

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



## Tables

**Table 1: Modified Lamb Weather Types (LWT, BSH) and Objective Classification (OWTC, DWD), Dezember 2014. Letters in LWT indicate from left to right: The classified weather type, cyclonality index ("A" or "C"), predominant wind direction at ground level, and gale index. Characters in OWTC indicate from left to right: Weather type number, predominant wind direction at 700 hPa, cyclonality ("A" or "Z") at 950 and 500 hPa, humidity index ("T" or "F"). LWT gale indices are printed in orange ("gale") und red ("severe gale") letters, OWTC wet weather types in blue letters. "NUL" indicates "no gale".**

| Date | LWT | OWTC |
|------|-----|------|
| 01/12/2014 | SE  A SE  NUL | 38 SO Z Z F |
| 02/12/2014 | NE  A NE  NUL | 21 XX Z A T |
| 03/12/2014 | A  A NE  NUL | 31 XX Z Z T |
| 04/12/2014 | A  A SE  NUL | **38 SO Z Z F** |
| 05/12/2014 | C  C SW  NUL | **9 SW A A F** |
| 06/12/2014 | A  A NW  NUL | **6 XX A A F** |
| 07/12/2014 | SW  A SW  NUL | 4 SW A A T |
| 08/12/2014 | NW  C NW  NUL | 15 NW A Z T |
| 09/12/2014 | SW  A SW  NUL | 11 XX A Z T |
| 10/12/2014 | **SW  C SW  SG** | 5 NW A A T |
| 11/12/2014 | **SW  C SW  G** | 15 NW A Z T |
| 12/12/2014 | **C  C SW  G** | **29 SW Z A F** |
| 13/12/2014 | NW  A NW  NUL | 4 SW A A T |
| 14/12/2014 | **SW  A SW  G** | **19 SW A Z F** |
| 15/12/2014 | SW  C SW  NUL | **9 SW A A F** |
| 16/12/2014 | NW  A NW  NUL | **19 SW A Z F** |
| 17/12/2014 | SW  C SW  NUL | **40 NW Z Z F** |
| 18/12/2014 | SW  C SW  NUL | **10 NW A A F** |
| 19/12/2014 | **NW  C NW  G** | **10 NW A A F** |
| 20/12/2014 | **NW  C NW  G** | 15 NW A Z T |
| 21/12/2014 | NW  A NW  NUL | 5 NW A A T |
| 22/12/2014 | **NW  A NW  G** | **10 NW A A F** |
| 23/12/2014 | **NW  A NW  G** | **10 NW A A F** |
| 24/12/2014 | C  C NW  NUL | **9 SW A A F** |
| 25/12/2014 | NW  C NW  NUL | 35 NW Z Z T |
| 26/12/2014 | A  A SW  NUL | 15 NW A Z T |
| 27/12/2014 | C  C SE  NUL | 31 XX Z Z T |
| 28/12/2014 | A  A NE  NUL | 2 NO A A T |
| 29/12/2014 | A  A NW  NUL | 35 NW Z Z T |
| 30/12/2014 | A  A NW  NUL | 5 NW A A T |
| 31/12/2014 | A  A SW  NUL | **7 NO A A F** |

**Figures**

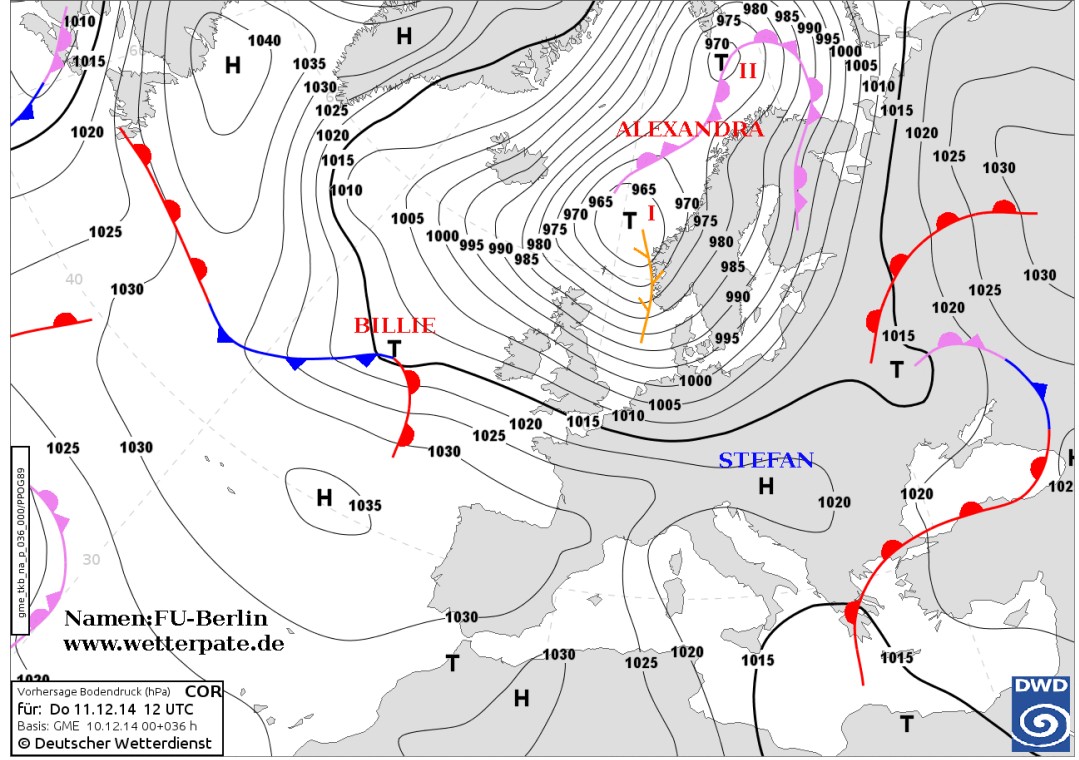

**Figure 1: Sea level pressure prediction for Thursday, 11 December 2014 showing the low pressure systems ALEXANDRA and BILLIE in short progression. Image credit: FU–Berlin, www.met.fu-berlin.de (last access: 20 February 2015).**



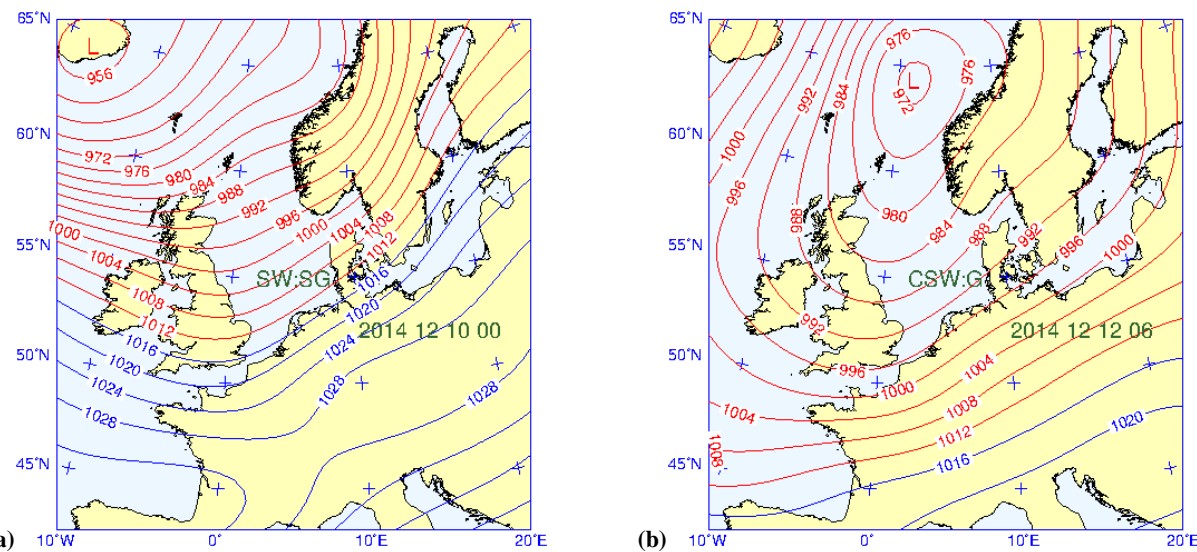

**Figure 2: Classification of the general weather situation (GWS) after the modified Lamb Weather Types used at BSH for (a) the low pressure systems ALEXANDRA (classification: South west (SW) with "severe gale" (SW)) and (b) BILLIE (classification: Cyclonal south west (CSW) with "gale" (G)). Image credit: P. Löwe (BSH Hamburg).**

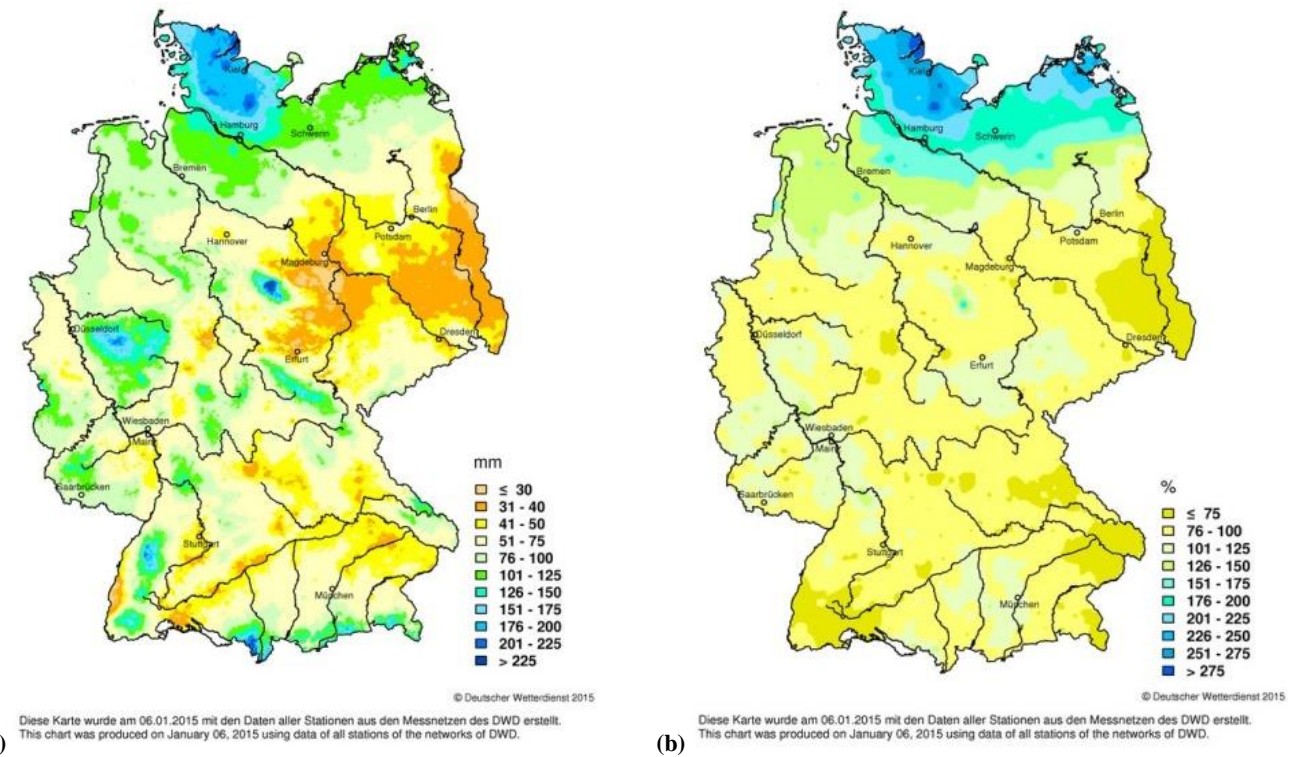

**Figure 3: (a) Precipitation sums in Germany in [mm], December 2014 and (b) its differences in [%] to the long term mean 1961–1990. Image credit: DWD, www.dwd.de (last access: 6 January 2015).**



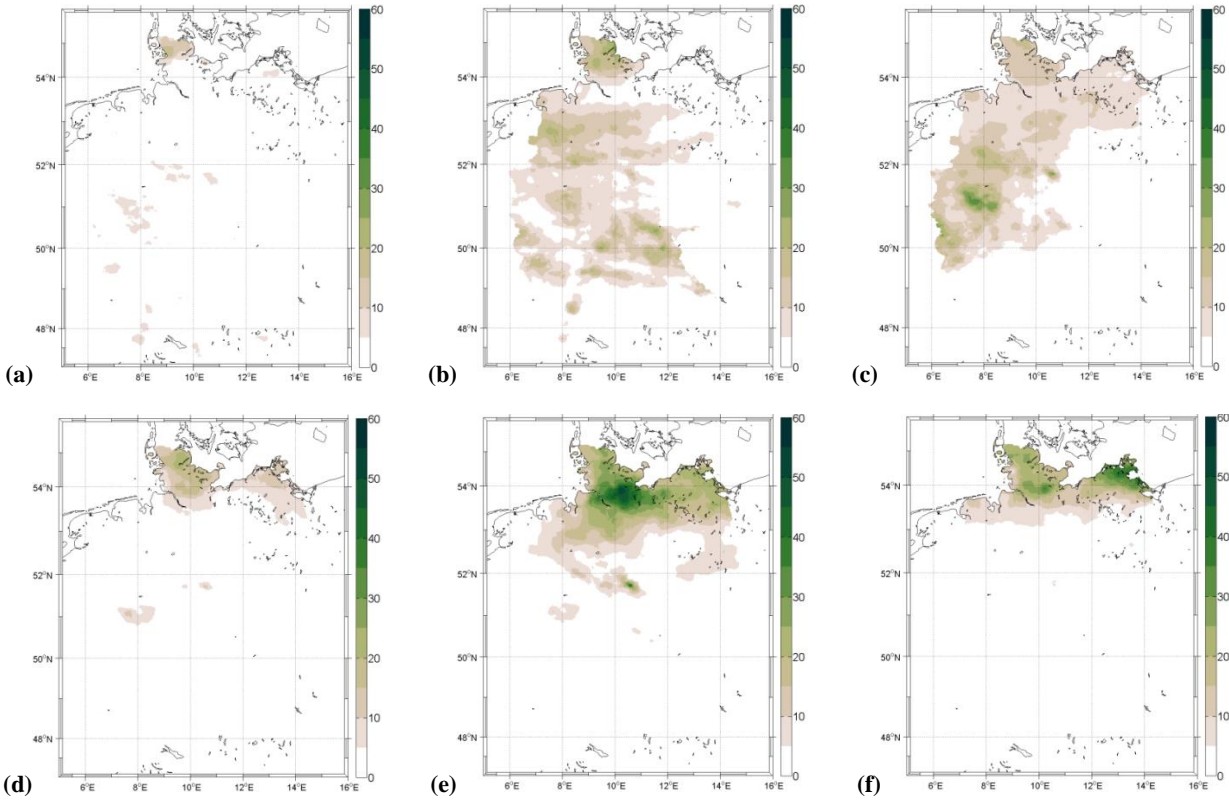

**Figure 4: Daily sums of REGNIE precipitation data in [mm] for the first event, 10–12 December 2014 (a–c), and the main precipitation event, 21–23 December 2014 (d–f).**

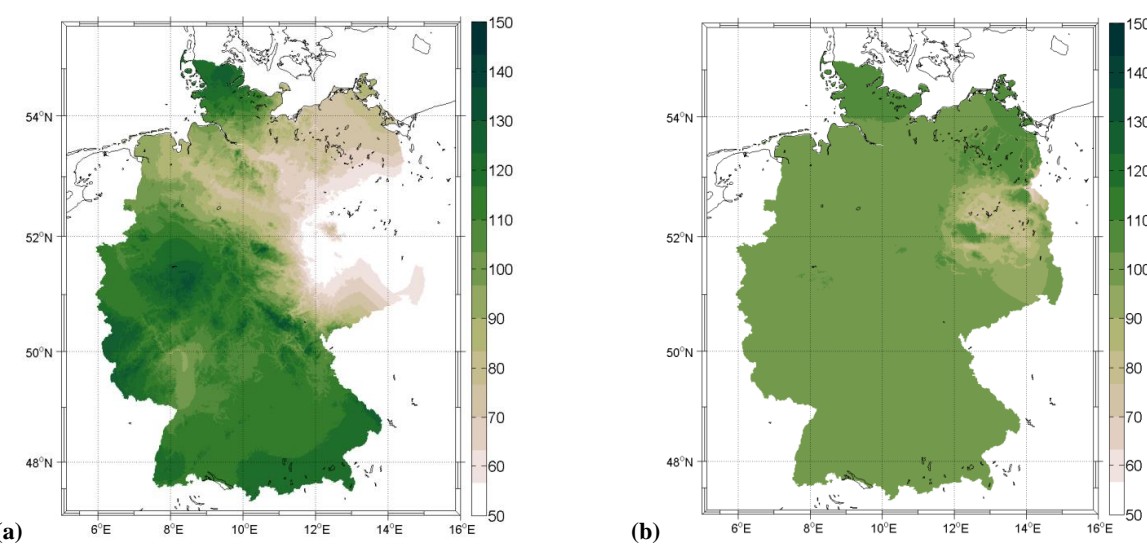

**Figure 5: (a) Soil moisture in [% nFK] for sandy loam soil and cultivation with sugar beets and (b) for loamy sand soil and cultivation with winter grain, 21 December 2014 (Model calculations by ZAMF, Braunschweig, Germany).**





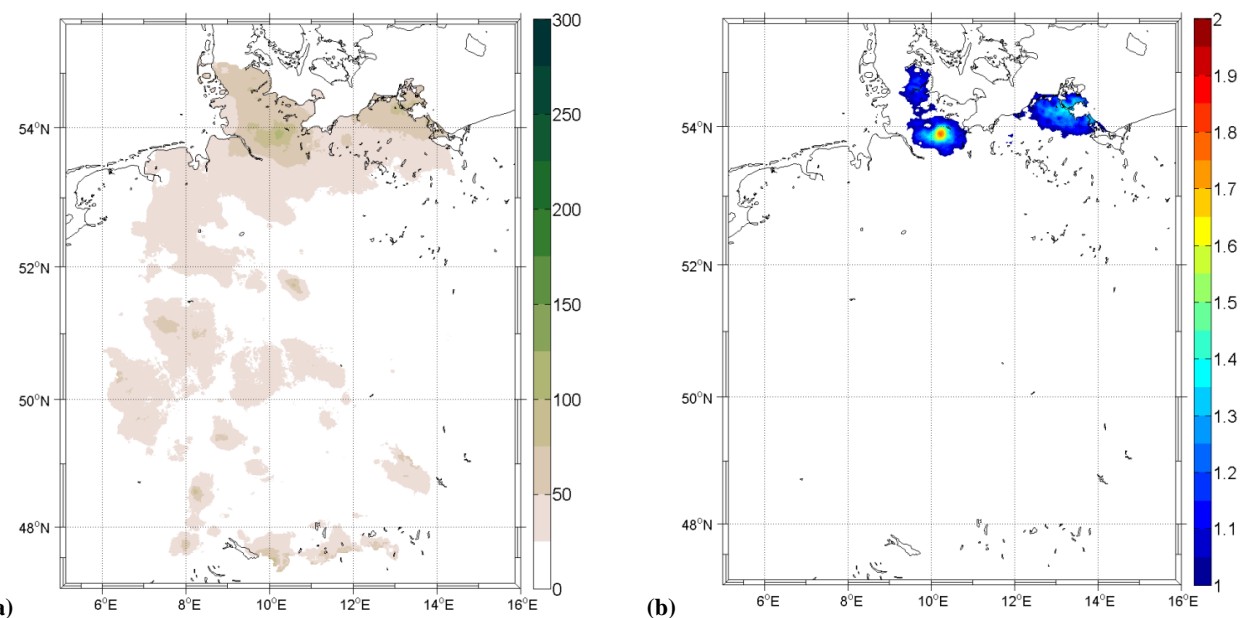

**Figure 6: (a) 3–day event precipitation (R3d) in [mm] and (b) its ratio to the 5y–return period (base period 1960–2009) for the December Flood 2014 in Schleswig–Holstein, Germany, calculated from REGNIE data following the method described in Schröter et al. (2015).**

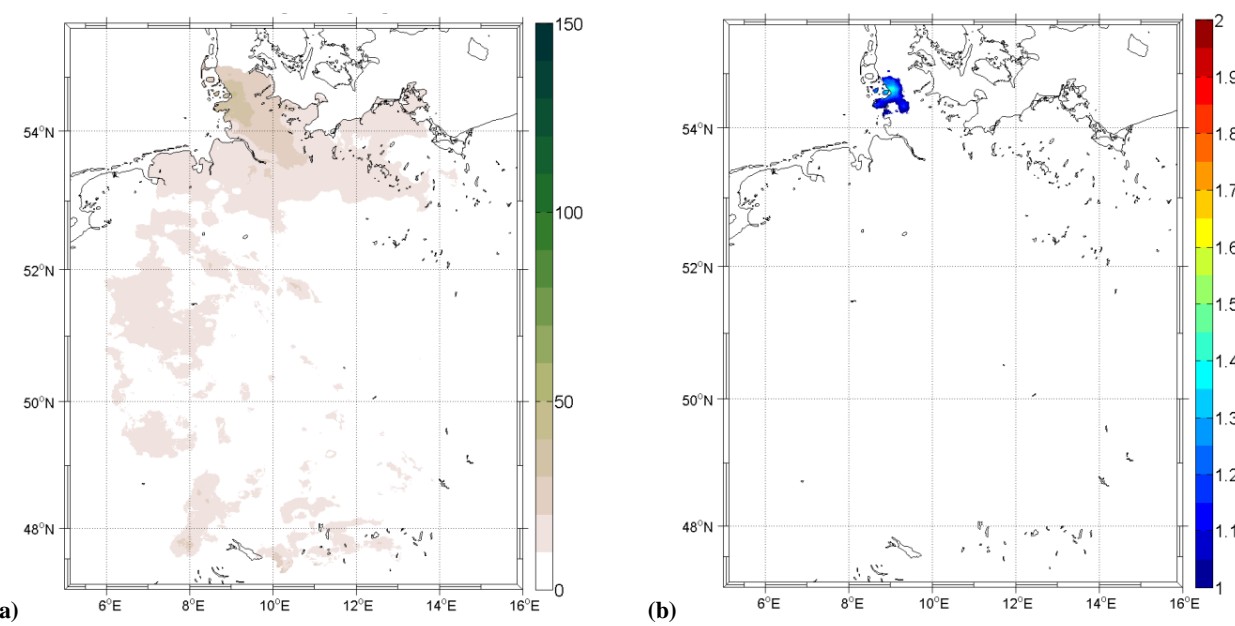

**Figure 7: See Figure 6, but for the antecedent precipitation index (API).**



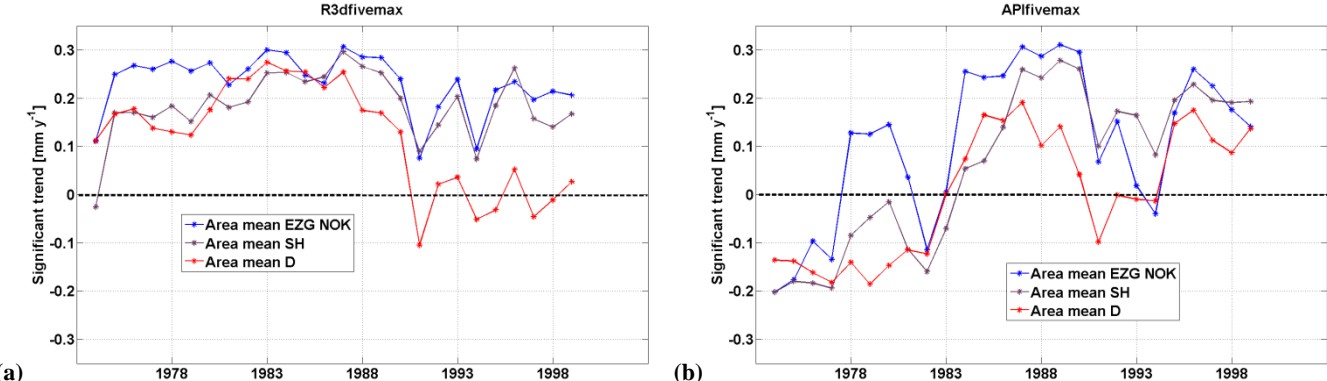

**Figure 8: Mean significant trends (95 % significance level) over 30–year running intervals from 1960–1989 to 19852–014 in [mm y⁻¹] for the five highest (a) 3–day event precipitation (R3dfivemax) and (b) antecedent precipitation indices (APIfivemax) per year for the Kiel Canal catchment (blue), Schleswig–Holstein (mauve) and all of Germany (red). The centre year of the respective 30–year time slices is marked on the x–axis.**

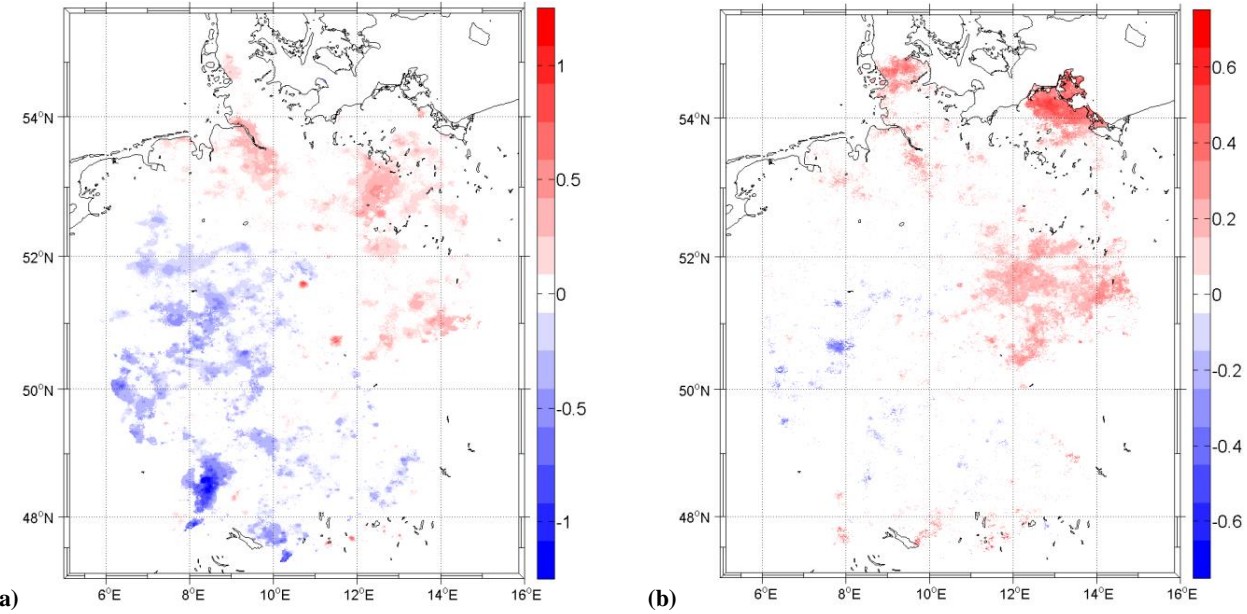

**Figure 9: Significant trends (95 % significance level) in [mm y⁻¹] for the five highest (a) 3–day event precipitation (R3dfivemax) and (b) antecedent precipitation indices (APIfivemax) per year in Germany, base period 1983–2012.**