# Peer review of "Evaluating the atmospheric drivers leading to the December Flood 2014 in Schleswig–Holstein, Germany"

_Earth System Dynamics, 2016_

## Referee Comment (RC1) · Anonymous Referee #1 · 30 Jan 2017

The author presents a study to estimate the atmospheric drivers of December flood 2014 in Schleswig–Holstein. Different types of classification methods and indices' combination (i.e. antecedent precipitation index & maximum 3 –day precipitation sum) as well as trends assessments were used to analyze spatial and temporal variability of flood events. The applied methodology seems to be technically sound.

General comments. The paper lacks a discussion on the basis and consistency of the chosen indices. There is a lack of hydrological information about the flood. References provided are not the best way to understand hydrological behavior during 21-23 of December 2014 in general. Conclusions present not relevant information about flood aftermath. A large part of the conclusion is devoted to the future plans. The language

sometimes is not fluent and the writing should be checked. These are comments (minor), which needs to be addressed before it is accepted for publication: P1L28 What is the reason for MIB to be mentioned? P1L29 Maybe it's better to use calendar dates P2L13 Not whole Europe, may be Northern Europe P6L15 Maybe it's better to provide some assessments of the REGNIE performing goodness then to refer on the figures of the reports P6L26 The end of the sentence "only that not differentiated" is unclear. What kind of differentiation? P7L6 To name a source of information P8L1 A verb (may be "need") is missing P8L3 "affecting the drainage of affected catchments" . Maybe it's better to re-write this part of the sentence to avoid unclearness. P8L20 Unmistakably is very strong form of certainty. May be it's better to use another word when talking about the future. P9L34 References proving additional meltwater runoff in the future are needed Maps should contain major catchment boundaries (including Kiel channel watershed)

---

## Referee Comment (RC2) · Anonymous Referee #2 · 2 Feb 2017

Referee comments to the manuscript

Evaluating the atmospheric drivers leading to the December Flood 2014 in Sch leswig-Holstein, Germany

presented by Nils H. Schade

In this article, atmospheric conditions were studied, which caused the severe flooding in Schleswig-Holstein in December 2014. The topic is interesting and important having a direct value for human activity. Two classifications of large-scale atmospheric circulation and two indices of precipitation and moisture conditions were used. The main disadvantage of the paper is its descriptive nature. A number of characteristics and

maps have been presented but their analysis, synthesis and discussion is lacking. The objectives, tasks and hypotheses of the study are not clearly formulated. I'll answer to the general questions of the journal and then I'll make my more detail comments and suggestions. 1. Does the paper address relevant scientific questions within the scope of ESD? Partly. 2. Does the paper present novel concepts, ideas, tools, or data? Some. 3. Are substantial conclusions reached? Partly. 4. Are the scientific methods and assumptions valid and clearly outlined? Yes. 5. Are the results sufficient to support the interpretations and conclusions? Yes. 6. Is the description of experiments and calculations sufficiently complete and precise to allow their reproduction by fellow scientists (traceability of results)? Yes. 7. Do the authors give proper credit to related work and clearly indicate their own new/original contribution? Yes. 8. Does the title clearly reflect the contents of the paper? Yes. 9. Does the abstract provide a concise and complete summary? Yes. 10. Is the overall presentation well structured and clear? Yes. 11. Is the language fluent and precise? Revision of the language is needed. 12.Are mathematical formulae, symbols, abbreviations, and units correctly defined and used? Yes. 13. Should any parts of the paper (text, formulae, figures, tables) be clarified, reduced, combined, or eliminated? No. 14. Are the number and quality of references appropriate? More or less, yes. 15. Is the amount and quality of supplementary material appropriate? Yes.

Remarks and suggestions 1. The term "westerly situation" widely used in this paper seems a bit strange for me. I think that "westerly circulation" is meant here. Classifications of general weather situations are more like circulation classifications (page 3 line 13). 2. Page 1 line 18. I prefer to use "precipitation event" instead of "event precipitation". 3. Page 1 line 27. There should be December, not Dezember. 4. Page 2 line 13-14. The sentence should be revised. Which physical conditions of the North Sea are the dominant factors? 5. Page 3 line 14. What does mean the abbreviation BSH? 6. I have a question how much the used circulation classifications were objective or subjective. It is written that the second classification was objective. But is the Jenkinson-Collison classification subjective? 7. Page 3 line 21. I am not sure which
term is used in English: "cyclonality" of "cyclonicity". Please, make clear it. 8. In the section 2.2 many data sources were listed. Which variables were used in this study, it was not indicated. 9. Trend analysis was not mentioned in the introduction. Why it was included into this study? Trends are not related to the 2014 flooding event. The significance of the trends is not estimated at all. Without it we cannot talk about trends. 10. It is not correct to express wind speed using the Beaufort scale. It will be better to do it using m/s. 11. I think that there are too many subchapters in the chapter 3. I recommend to use two hierarchic levels, not three. 12. Page 5, line 24. There is written that the LWT seems more appropriate. How much is this statement justified? On which facts is it based? 13. A misunderstanding is related to the title 3.1.4 Gauge data. Are the data from rain gauges? In fact, there is information about water level measurements. Gauge data were not described in the section of data and methods. 14. Page 8 lines 3-4. This sentence was not understandable for me. 15. It was difficult to understand the use of severity indices. What they show and how they could be compared? 16. The main results of the study are not clearly and shortly concluded
* * *

---

## Author Comment (AC1) · 17 Feb 2017

*esd-2016-73-RC1*
*Schade, N.H.*
*"Evaluating the atmospheric drivers leading to the December Flood 2014 in Schleswig-Holstein, Germany"*

*Content:*
*The author presents a study to estimate the atmospheric drivers of December flood 2014 in Schleswig–Holstein. Different types of classification methods and indices' combination (i.e. antecedent precipitation index & maximum 3 –day precipitation sum) as well as trends assessments were used to analyze spatial and temporal variability of flood events. The applied methodology seems to be technically sound.*

*General comments:*
*The paper lacks a discussion on the basis and consistency of the chosen indices. There is a lack of hydrological information about the flood. References provided are not the best way to understand hydrological behavior during 21-23 of December 2014 in general. Conclusions present not relevant information about flood aftermath. A large part of the conclusion is devoted to the future plans. The language sometimes is not fluent and the writing should be checked.*

**Response to general comments**: At first, I would like to thank the anonymous referee #1 for the helpful criticism to improve this manuscript.

A discussion on the basis and consistency of the indices will be included in the revised manuscript. However, since it is the first attempt to scale the indices down to the regional level, there is not much information concerning consistency yet. This is in fact part of ongoing investigations within the Expertennetzwerk.

Further, it should be pointed out that this paper was not intended to describe the hydrology in detail since that investigation was already performed by the LLUR/LKN-SH. The focus rather lies on the meteorological information leading to this regionally confined flood event and how well the indices for event precipitation (R3d) and antecedent precipitation (API) can describe the onset of this flood. It was already shown by Schröter et al. (2015) that the indices are well suited to describe the onset of nation-wide flood events. I fully agree, however, that referring to another investigation is not the best way to understand the hydrology. The revised manuscript will include the referred picture (with kind permission of the LLUR-SH) showing almost perfect agreement between regions where API and R3d are exceeding their respective 5-year return periods and inland gauges exceeding highest water levels. Further, a passage will be added concerning gauge data to describe the regional concurrence, as well as in the "Data and Methodology" chapter.

The concluding chapter will be removed of non-relevant information and rewritten according to the focus of the paper. Nevertheless, this work is a starting point for further investigations (or future plans) that will include more in depth analyses and, hopefully, provide important information on how climate change is affecting the indices used in this paper. As mentioned before, this is the first attempt to scale the indices down to the regional level and results seem

promising that they might be useful for getting a first glimpse into future changes without the need to run hydrological models (which of course will be the next step for flood protection, adaptation measures, etc.). Therefore, the revised manuscript will include a separate outlook chapter.

Finally, I would like the revised manuscript to be proofread by the editorial office to improve the language.

*These are comments (minor), which needs to be addressed before it is accepted for publication:*

*1.) P1L28 What is the reason for MIB to be mentioned?*

**Response**: MIB was mentioned to highlight the various responses to the same cause: Persistent westerly circulation. Also, it was intended to tie the analysis more into to the investigations performed within Baltic Earth already and to pique the interest of the community in our work and the work to come. I agree, however, it does not further improve the paper and its purpose. This part would be removed in the revised manuscript, if the editor decides it is of no further interest to the community.

*2.) P1L29 Maybe it's better to use calendar dates*

**Response**: Calendar dates will be used in the revised manuscript.

*3.) P2L13 Not whole Europe, may be Northern Europe*

**Response**: The revised manuscript will be changed accordingly

*4.) P6L15 Maybe it's better to provide some assessments of the REGNIE performing goodness then to refer on the figures of the reports*

**Response**: A Figure and a passage comparing REGNIE to selected observed station data can be included in the revised manuscript.

*5.) P6L26 The end of the sentence "only that not differentiated" is unclear. What kind of differentiation?*

**Response**: Soil moisture for sand soil shows the same structure, i.e. almost the same values all over Schleswig-Holstein whereas loam soil is clearly wetter in the Northern parts and drier in the South. The respective passage will be rephrased in the revised manuscript.

*6.) P7L6 To name a source of information*

**Response**: The source of information will be added in the revised manuscript.

*7.) P8L1 A verb (may be "need") is missing*

**Response**: Yes, indeed. It will be added in the revised manuscript

*8.) P8L3 "affecting the drainage of affected catchments". Maybe it's better to re-write this part of the sentence to avoid unclearness.*

**Response**: The respective passage will be rephrased in the revised manuscript to avoid unclearness.

*9.) P8L20 Unmistakably is very strong form of certainty. May be it's better to use another word when talking about the future.*

**Response**: You're right; those strong forms better be avoided. Thanks for pointing it out!

*10.) P9L34 References proving additional meltwater runoff in the future are needed*

**Response**: Admittedly, it was an assumption on my part. References will be included in the revised manuscript or the passage will be rephrased/removed in case of missing references.

*11.) Maps should contain major catchment boundaries (including Kiel channel watershed)*

**Response**: All respective figures in the revised manuscript will include boundaries of Schleswig-Holstein and the Kiel Canal catchment to make the figures easier to understand.

---

## Author Response (AR1)

To the editor

Dear Dr. Rutgersson,
please find enclosed my revised manuscript, Number: esd-2016-73

Author: Dr. Nils H. Schade
Title: Evaluating the atmospheric drivers leading to the December Flood 2014 in Schleswig–Holstein, Germany.

The text has been thoroughly revised according to the criticism and suggestions for changes of the reviewers.

A point-by-point response to the review items and the marked-up manuscript is attached. As you can see from this attachment the criticisms and suggestions have been taken into account. I would like to thank the reviewers for their helpful criticism. Their feedback was important to clarify the main intention of my work.

The following major changes have been made:

1.) Hydrological information about the flood has been added to the manuscript including a new Figure 9 showing gauge exceeding highest high water levels.

2.) A more detailed assessment of the REGNIE performing goodness has been added including a new Figure 5 comparing REGNIE to DWD station data.

3.) Hierarchic levels have been reduced.

4.) The chapters concerning severity indices have been removed since it does not further improve the manuscript. Accordingly, Figures 7b/8b now show return periods instead of index values.

5.) The summary and conclusion chapter has been rewritten to shortly recap the findings and a separate outlook chapter has been introduced.

6.) Administrative and catchment boundaries have been included in Figures 4, 6, 7, 8, and 11.

Finally, I would like my manuscript to be proofread by the editorial office to improve the language. I am hoping the manuscript is then in a more appropriate shape for publication.

Kind regards,

Dr. Nils H. Schade

*esd-2016-73-RC1*
*Schade, N.H.*
*"Evaluating the atmospheric drivers leading to the December Flood 2014 in Schleswig-Holstein, Germany"*

*Content:*
*The author presents a study to estimate the atmospheric drivers of December flood 2014 in Schleswig–Holstein. Different types of classification methods and indices' combination (i.e. antecedent precipitation index & maximum 3 –day precipitation sum) as well as trends assessments were used to analyze spatial and temporal variability of flood events. The applied methodology seems to be technically sound.*

*General comments:*
*The paper lacks a discussion on the basis and consistency of the chosen indices. There is a lack of hydrological information about the flood. References provided are not the best way to understand hydrological behavior during 21-23 of December 2014 in general. Conclusions present not relevant information about flood aftermath. A large part of the conclusion is devoted to the future plans. The language sometimes is not fluent and the writing should be checked.*

**Response to general comments**: At first, I would like to thank the anonymous referee #1 for the helpful criticism to improve this manuscript.

Since it is the first attempt to scale the indices down to the regional level, there is not much information concerning consistency yet. This is in fact part of ongoing investigations in the NOK catchment area. Preliminary results show that R3d and API seem to be promising indicators/predictors to describe problematic situations in the channel's operational routine. However, since many other influencing factors like sea level rise, wind surge, locking of ships, dewatering, ferry trafficking, etc. are involved, pin pointing the respective factors to one single event is difficult, and therefore, it is not possible to give an accurate estimation of the consistency for regional investigations at this point yet.

Further, it should be pointed out that this paper was not intended to describe the hydrology in detail since that investigation was already performed by the LLUR/LKN-SH. The focus rather lies on the meteorological information leading to this regionally confined flood event and how well the indices for event precipitation (R3d) and antecedent precipitation (API) can describe the onset of this flood. It was already shown by Schröter et al. (2015) that the indices are well suited to describe the onset of nation-wide flood events.

I fully agree, however, that referring to another investigation is not the best way to understand the hydrology. The revised manuscript now includes the referred picture (with kind permission of the LLUR-SH) showing almost perfect agreement between regions where API and R3d are exceeding their respective 5-year return periods and inland gauges exceeding highest water levels. Further, a passage has been added concerning gauge data to describe the regional concurrence, as well as in the "Data and Methodology" chapter.

The concluding chapter has been removed of non-relevant information. Nevertheless, this work

is a starting point for further investigations (or future plans) that will include more in depth analyses and, hopefully, provide important information on how climate change is affecting the indices used in this paper. As mentioned before, this is the first attempt to scale the indices down to the regional level and results seem promising that they might be useful for getting a first glimpse into future changes without the need to run hydrological models (which of course will be the next step for flood protection, adaptation measures, etc.). Therefore, the revised manuscript now includes a separate outlook chapter.

Finally, I would like the revised manuscript to be proofread by the editorial office to improve the language.

*These are comments (minor), which needs to be addressed before it is accepted for publication:*

*1.) P1L28 What is the reason for MIB to be mentioned?*

**Response**: MIB was mentioned to highlight the various responses to the same cause: Persistent westerly circulation. Also, it was intended to tie the analysis more into to the investigations performed within Baltic Earth already and to pique the interest of the community in our work and the work to come. I agree, however, it does not further improve the paper and its purpose. This part will be removed, if the editor decides it is of no further interest to the community.

*2.) P1L29 Maybe it's better to use calendar dates*

**Response**: Calendar dates are now used in the revised manuscript.

*3.) P2L13 Not whole Europe, may be Northern Europe*

**Response**: The revised manuscript has been changed accordingly

*4.) P6L15 Maybe it's better to provide some assessments of the REGNIE performing goodness then to refer on the figures of the reports*

**Response**: A Figure and a passage comparing REGNIE to selected observed station data is included in the revised manuscript.

*5.) P6L26 The end of the sentence "only that not differentiated" is unclear. What kind of differentiation?*

**Response**: Soil moisture for sand soil shows the same structure, i.e. almost the same values all over Schleswig-Holstein whereas loam soil is clearly wetter in the Northern parts and drier in the South. The respective passage has been removed in the revised manuscript to avoid unclearness.

*6.) P7L6 To name a source of information*

**Response**: The source of information has been added in the revised manuscript.

*7.) P8L1 A verb (may be "need") is missing*

**Response**: Yes, indeed. It has been added in the revised manuscript

*8.) P8L3 "affecting the drainage of affected catchments". Maybe it's better to re-write this part of the sentence to avoid unclearness.*

**Response**: The respective passage has been rephrased in the revised manuscript to avoid unclearness.

*9.) P8L20 Unmistakably is very strong form of certainty. May be it's better to use another word when talking about the future.*

**Response**: You're right; those strong forms better be avoided. Thanks for pointing it out!

*10.) P9L34 References proving additional meltwater runoff in the future are needed*

**Response**: Admittedly, it was an assumption on my part. The passage has been removed in the revised manuscript.

*11.) Maps should contain major catchment boundaries (including Kiel channel watershed)*

**Response**: All respective figures in the revised manuscript now include boundaries of Schleswig-Holstein and the Kiel Canal catchment to make the figures easier to understand.

*esd-2016-73-RC2*
*Schade, N.H.*
*"Evaluating the atmospheric drivers leading to the December Flood 2014 in Schleswig-Holstein, Germany"*

*Content:*
*In this article, atmospheric conditions were studied, which caused the severe flooding in Schleswig-Holstein in December 2014. The topic is interesting and important having a direct value for human activity. Two classifications of large-scale atmospheric circulation and two indices of precipitation and moisture conditions were used.*

*General comments:*
*The main disadvantage of the paper is its descriptive nature. A number of characteristics and maps have been presented but their analysis, synthesis and discussion is lacking. The objectives, tasks and hypotheses of the study are not clearly formulated. I'll answer to the general questions of the journal and then I'll make my more detail comments and suggestions. 1. Does the paper address relevant scientific questions within the scope of ESD? Partly. 2. Does the paper present novel concepts, ideas, tools, or data? Some. 3. Are substantial conclusions reached? Partly. 4. Are the scientific methods and assumptions valid and clearly outlined? Yes. 5. Are the results sufficient to support the interpretations and conclusions? Yes. 6. Is the description of experiments and calculations sufficiently complete and precise to allow their reproduction by fellow scientists (traceability of results)? Yes. 7. Do the authors give proper credit to related work and clearly indicate their own new/original contribution? Yes. 8. Does the title clearly reflect the contents of the paper? Yes. 9. Does the abstract provide a concise and complete summary? Yes. 10. Is the overall presentation well structured and clear? Yes. 11. Is the language fluent and precise? Revision of the language is needed. 12.Are mathematical formulae, symbols, abbreviations, and units correctly defined and used? Yes. 13. Should any parts of the paper (text, formulae, figures, tables) be clarified, reduced, combined, or eliminated? No. 14. Are the number and quality of references appropriate? More or less, yes. 15. Is the amount and quality of supplementary material appropriate? Yes.*

**Response to general comments**: At first, I would like to thank the anonymous referee #2 for the helpful criticism to improve this manuscript.

The revised manuscript has been rewritten with more emphasize on analyses, synthesis and discussion to state the objectives, tasks and hypotheses of the study more clearly (as also pointed out in the response to RC1).

Further, I would like the revised manuscript to be proofread by the editorial office to improve the language since both referees are pointing out language revision.

*Remarks and suggestions*

*1.) The term "westerly situation" widely used in this paper seems a bit strange for me. I think that "westerly circulation" is meant here. Classifications of general weather situations are more like circulation classifications (page 3 line 13).*

**Response**: The term "westerly situation" has been changed in the revised manuscript to "westerly circulation"

*2.) Page 1 line 18. I prefer to use "precipitation event" instead of "event precipitation".*

**Response**: The term "event precipitation" was defined by Schröter et al. (2015) as the highest 3-day precipitation sum at the onset of the flood. For consistency reasons I would prefer to keep it.

*3.) Page 1 line 27. There should be December, not Dezember.*

**Response**: Indeed!

*4.) Page 2 line 13-14. The sentence should be revised. Which physical conditions of the North Sea are the dominant factors?*

**Response**: The sentence has been rewritten.

*5.) Page 3 line 14. What does mean the abbreviation BSH?*

**Response**: "Federal Maritime and Hydrographic Agency" has been included in the revised manuscript.

*6.) I have a question how much the used circulation classifications were objective or subjective. It is written that the second classification was objective. But is the Jenkinson-Collison classification subjective?*

**Response**: The Lamb Weather Types (LWT) in the original form are indeed subjective. Jenkinson and Collison developed an automated system to "objectify" LWT, allowing classification based on sea level pressure data solely. Therefore, the Jenkinson-Collison classification is objective as well. It is now stated in the revised manuscript.

*7.) Page 3 line 21. I am not sure which term is used in English: "cyclonality" of "cyclonicity". Please, make clear it.*

**Response**: "Cyclonality" is the correct term.

*8.) In the section 2.2 many data sources were listed. Which variables were used in this study, it was not indicated.*

**Response**: The REGNIE dataset includes only daily precipitation sums (on a 1 km by 1 km grid) which were used in this study to calculate the precipitation indices. The chapter has been rewritten in the revised manuscript to clarify.

*9.) Trend analysis was not mentioned in the introduction. Why it was included into this study? Trends are not related to the 2014 flooding event. The significance of the trends is not estimated at all. Without it we cannot talk about trends.*

**Response**: "trend analysis of the indices investigated" was mentioned on page 3 line 9, but I admit, it can easily be overlooked. The introduction has been rewritten to clarify that trends for the precipitation indices were included to point at potential future problems that may come with increased antecedent precipitation (-> higher soil moisture -> higher chance of flooding due to persistent precipitation) and increased event precipitation (-> higher chance of flooding due to higher precipitation sum). Concerning significance, only present significant trends (Mann-Kendall Test) are presented in the analyses, see chapter 2.3, page 5. However, supplementary information including figures showing maps of the $p\_value \leq 0.05$ could be added to the revised manuscript if the editor decides it is necessary.

*10.) It is not correct to express wind speed using the Beaufort scale. It will be better to do it using m/s.*

**Response**: Well, I would not say it is incorrect to use the Beaufort scale, since wind speed observations over sea have been estimated in Beaufort for a long time and, in fact, when comparing those observations with today's measurements, it is always recommended to use the Beaufort scale. But I admit that it is better to use m/s in this context.

*11.) I think that there are too many subchapters in the chapter 3. I recommend to use two hierarchic levels, not three.*

**Response**: The hierarchic levels haven been changed in the revised

manuscript.

*12.) Page 5, line 24. There is written that the LWT seems more appropriate. How much is this statement justified? On which facts is it based?*

**Response**: Actually, it is due to the fact that LWT is centred close to the area of interest and offers the slightly better suited general weather situation for this specific case. Further, OWTC misses wet days during the first precipitation event. The passage has been rewritten in the revised manuscript.

*13.) A misunderstanding is related to the title 3.1.4 Gauge data. Are the data from rain gauges? In fact, there is information about water level measurements. Gauge data were not described in the section of data and methods.*

**Response**: No, these are not rain gauges. As pointed out in the response to RC1, there are additional passages in the revised manuscript including information about gauge data from the report of the LKN-SH and LLUR-SH (2015).

*14.) Page 8 lines 3-4. This sentence was not understandable for me.*

**Response**: The passage has been rewritten in the revised manuscript.

*15.) It was difficult to understand the use of severity indices. What they show and how they could be compared?*

**Response**: The severity indices are measures to compare flood events in their extent and extremeness. I admit chapter 3.2.3 does not really improve the manuscript in this regard. The chapter has been removed in the revised manuscript, together with passages in the "Data and Methodology" chapter.

*16.) The main results of the study are not clearly and shortly concluded*

**Response**: As pointed out in the response to RC1, the concluding remarks have been extensively rewritten according to the focus of the paper.

[revised manuscript text omitted]